# Effects of Whole-Body Vibration Exercises on Parameters Related to the Sleep Quality in Metabolic Syndrome Individuals: A Clinical Trial Study

**Claudia Figueiredo Azeredo** [1,2], **Patrícia de Castro de Paiva** [1,2], **Leandro Azeredo** [3],
**Aline Reis da Silva** [1,2], **Arlete Francisca-Santos** [2,4], **Laisa Liane Paineiras-Domingos** [2,4,5],
**Adriana Lírio Pereira da Silva** [2], **Camila Leite Bernardes-Oliveira** [1,5], **Juliana Pessanha-Freitas** [2],
**Márcia Cristina Moura-Fernandes** [2,6], **Rubens Guimarães Mendonça** [2], **José Alexandre Bachur** [7],
**Ygor Teixeira-Silva** [2,5], **Eloá Moreira-Marconi** [2,6], **Eliane de Oliveira Guedes-Aguiar** [2,4,5],
**Bruno Bessa Monteiro de Oliveira** [2,4], **Mário Fritsch Neves** [5], **Luiz Felipe Ferreira-Souza** [2],
**Vinicius Layter Xavier** [8], **Daniel Lago Borges** [9], **Ana Cristina Lacerda** [10],
**Vanessa Amaral Mendonça** [10], **Anelise Sonza** [11], **Redha Taiar** [12,*], **Alessandro Sartorio** [13],
**Mario Bernardo-Filho** [2] **and Danúbia da Cunha de Sá-Caputo** [2,4,5]

1 Programa de Pós-graduação em Saúde, Medicina Laboratorial e Tecnologia Forense, Universidade do Estado do Rio de Janeiro, Rio de Janeiro, RJ 20950-003, Brazil; figueiredo.claudiaf@gmail.com (C.F.A.); patriciadecastrodepaiva@gmail.com (P.d.C.d.P.); fisioalinereis@gmail.com (A.R.d.S.); camilabernardes@hotmail.com (C.L.B.-O.)

2 Laboratório de Vibrações Mecânicas e Práticas Integrativas, Departamento de Biofísica e Biometria, Instituto de Biologia Roberto Alcântara Gomes e Policlínica Piquet Carneiro, Universidade do Estado do Rio de Janeiro, Rio de Janeiro, RJ 20950-003, Brazil; fisioarlete@gmail.com (A.F.-S.); laisanit@gmail.com (L.L.P.-D.); adrianaliriolavimpi@gmail.com (A.L.P.d.S.); ju.freitas.fisio@gmail.com (J.P.-F.); marciafernandesfisio@gmail.com (M.C.M.-F.); rubens_fisio@hotmail.com (R.G.M.); silvarogy@hotmail.com (Y.T.-S.); eloamarconi@gmail.com (E.M.-M.); ellianeguedes@gmail.com (E.d.O.G.-A.); bessa.oliveira@gmail.com (B.B.M.d.O.); lumadaragu@gmail.com (L.F.F.-S.); bernardofilhom@gmail.com (M.B.-F.); dradanubia@gmail.com (D.d.C.d.S.-C.)

3 Departamento de Fisioterapia, Hospital Central da Polícia Militar, Rio de Janeiro, RJ 20211-270, Brazil; azeredo.lm@gmail.com

4 Faculdade Bezerra de Araújo, Rio de Janeiro, RJ 23052-180, Brazil

5 Programa de Pós-Graduação em Ciências Médicas, Faculdade de Ciências Médicas, Universidade do Estado do Rio de Janeiro, Rio de Janeiro, RJ 20551-030, Brazil; mariofneves@gmail.com

6 Programa de Pós-Graduação em Fisiopatologia Clínica e Experimental, Instituto de Biologia Roberto Alcântara Gomes, Universidade do Estado do Rio de Janeiro, Rio de Janeiro, RJ 20550-900, Brazil

7 Cursos de Medicina e Fisioterapia da Universidade de Franca, Franca, SP 14401-426, Brazil; jabachur@hotmail.com

8 Departamento de Estatística, Instituto de Matemática e Estatística, Universidade do Estado do Rio de Janeiro, Rio de Janeiro, RJ 20550-900, Brazil; viniciuslx@ime.uerj.br

9 Hospital Universitário, Universidade Federal do Maranhão, São Luís do Maranhão, MA 65080-805, Brazil; dlagofisio83@hotmail.com

10 Faculdade de Ciências Biológicas e da Saúde, Universidade Federal dos Vales do Jequitinhonha e Mucuri, Diamantina, MG 39100-000, Brazil; lacerdaacr@gmail.com (A.C.L.); vaafisio@hotmail.com (V.A.M.)

11 Programa de Pós-graduação em Fisioterapia, Universidade do Estado de Santa Catarina, Florianópolis, SC 88080-350, Brazil; anelise.sonza@gmail.com

12 GRESPI, Université de Reims, 51100 Reims, France

13 Istituto Auxologico Italiano, IRCCS, Experimental Laboratory for Auxo-Endocrinological Research, Division of Metabolic Diseases & Auxology, 28824 Milan and Verbania, Italy; sartorio@auxologico.it

* Correspondence: redha.taiar@univ-reims.fr

**Abstract:** Metabolic syndrome (MetS) is an undesirable clinical condition with physiological, biochemical, clinical, and metabolic factors that contribute to increased cardiovascular risks (CR). A poor sleep quality might be found in obese and MetS individuals. Whole-body vibration (WBV) exercise has been used on the management of MetS individuals. This clinical trial investigated the effect of WBV exercise on parameters related to the sleep quality in MetS individuals. After randomization, nine individuals (seven women and two men) were exposed to a fixed frequency (FF) and ten individuals (eight women and two men) were exposed to a variable frequency (VF). Both groups performed the protocol twice a week, for 6 weeks. All of the evaluations were performed before the first and after the last sessions. Anthropometric and cardiovascular parameters were measured before and after the 6-week intervention. Pittsburgh Sleep Quality Index (PSQI), Epworth Sleepiness Scale (ESS), and Berlin Questionnaire were also used to evaluate the quality of the sleep. A significant ($p \leq 0.05$) reduction of the waist circumference in the VFG and an increase of the heart rate were found in the FFG and VFG group. The score of the PSQI of the both groups decreased significantly ($p = 0.01$). The score of the ESS decreased ($p = 0.04$) only in the VF group. The scores of the Berlin Questionnaire were not altered in both groups. In conclusion, WBV intervention was capable in interfering with physiological mechanisms with effects on the WC and HR, leading to the improvement of the quality of sleep in MetS individuals. WBV exercise might be an important clinical intervention to the management of some factors associated with poor quality of sleep (FFG and VFG) and in the daytime sleepiness in MetS individuals with variable frequencies (5–16 Hz) (VFG).

**Keywords:** metabolic syndrome; sleep quality; whole-body vibration exercise; Pittsburgh Sleep Quality Index; Epworth Sleepiness Scale; Berlin Questionnaire

---

## 1. Introduction

Metabolic syndrome (MetS) is an undesirable clinical condition with physiological, biochemical, clinical, and metabolic factors, including alterations on the level of the lipids on the plasma, arterial hypertension, central adiposity, insulin resistance (IR), and hyperglycemia. These conditions strongly contribute to increased cardiovascular risks (CR) [1], and they are interconnected by pathophysiological basis in low-grade chronic inflammation, increase of the risk of type 2 diabetes mellitus (T2DM), and all-cause mortality [2]. MetS is also related to the increased deposit of central adiposity and ectopic fat infiltration (muscle and liver) associated with overeating and/or sedentary lifestyles. These conditions are related to various deleterious consequences in late life [3,4].

Lian et al. [5], in a systematic review and meta-analysis, reported that the overall sleep quality as well as sleep complaints have significant positive associations with MetS. Massar et al. [6] and Tsai et al. [7] pointed out that this condition of the sleep is linked with cardiovascular disease. Massar et al. [6] reported in individuals with poor habitual sleep efficiency during the week before stress induction (Trier Social Stress Test) responded with higher stress-related elevations of blood pressure and cortisol levels as compared to subjects with high sleep efficiency. This relationship between poor sleep efficiency and elevated blood pressure persisted during the post-stress recovery period. In addition, poor sleep health is also associated with MetS and mental illness [8,9]. Iftikhar et al. [10], in a meta-analysis reported that some investigations demonstrated the association between sleep quality and MetS. It was verified that the poor sleep quality characterized by sleep fragmentation was related to impaired glucose metabolism, independent of sleep duration [11], however, the studies of the meta-analysis analyzed had not reached a consensus. Mesas et al. [12] reported that the difficulty falling asleep is associated with MetS and, in particular, with high blood pressure, and this association would be independent of sleep duration and would be not due to lifestyles related to poor sleep.

Ying et al. [5] reported in a systematic review and meta-analysis that the overall sleep quality, as well as sleep complaints have significant positive associations with MetS. Moreover, it is estimated [13,14] that 50% to 60% of obese and MetS individuals can present obstructive sleep apnea (OSA).

Some tools have been used to verify parameters (such as sleep duration, sleep inertia, sleep latency, snoring, disorientation or confusion during sleep, and daytime sleepiness) related to the quality of the sleep, such as the Pittsburgh Sleep Quality Index (PSQI) [15], Epworth Sleepiness Scale (ESS) [16] and the Berlin Questionnaire [17]. Moreover, the neck circumference (NC) [16] and waist circumference (WC) [18,19] have also been used.

There is moderate evidence supporting the use of physical exercise (PE) programs to counter MetS, although the optimal dose and type of PE is not well established. It is the main challenge for health care professionals to persuade individuals to have adherence to perform PE to the prevention and management of MetS [20]. In this case, it is relevant to consider the whole-body vibration (WBV) exercise, a modality of PE, an option. WBV exercise has been used as a clinical intervention in individuals with different clinical disorders, including MetS [21–26] and it can be considered a feasible, safe, and low-cost technique [27].

WBV exercise is generated in an individual who is on the base of a vibrating platform. It produces a mechanical vibration (MV) that is transmitted to the body inducing muscle contractions, with physiological responses like those produced by other types of PE, such as aerobic conditioning and strength training [28–30]. Biomechanical parameters, such as the frequency ($f$) and the peak-to-peak displacement (PPD) of the MV must be considered in the WBV exercise protocols [26,31,32].

As it is suggested that the sleep quality plays an important role in development of MetS [5] and putting together all of the previous considerations, to our knowledge, this is a first work aiming to study the effect of WBV exercise on the sleep quality of MetS individuals using the PSQI, ESS, and Berlin Questionnaire. It is hypothesized that the WBV exercise would be able in improving the sleep quality of MetS individuals.

## 2. Materials and Methods

### 2.1. Individuals

In this cross-sectional and randomized study, 31 subjects were recruited and 19 performed the protocol (58.79 ± 12.55 years old, 1.62 ± 0.09 m height, 86.27 ± 15.03 kg body mass). The recruitment of the participants occurred from January 2018 to January 2019, in the *Hospital Universitário Pedro Ernesto* (HUPE), in the *Universidade do Estado do Rio de Janeiro* (UERJ). The WBV protocol was performed in the *Laboratório de Vibrações Mecânicas e Práticas Integrativas* (LAVIMPI).

This project was approved by the Research Ethics Committee of HUPE-UERJ with the number CAAE 54981315.6.0000.5259, the registry in the Brazilian Registry of Clinical Trials (ReBEC) with the number RBR 2bghmh and UTN: U1111-1181-1177. The participants of both groups signed a consent form.

Consolidated Standards of Reporting Trials (CONSORT) was used to report all of the different steps of the interventions utilized in this work [33].

In the randomization, a blinded envelope was used for the cards with the name of the groups: fixed frequency group (FFG) (control group) or variable frequency group (VFG) (WBVE group).

The inclusion criteria were individuals of both sexes, over 18 years of age, with MetS according to the International Diabetes Federation.

The exclusion criteria were: (i) individuals with high blood pressure (BP) levels (systolic blood pressure (SBP) ≥ 180 mmHg and diastolic blood pressure (DBP) ≥ 110 mmHg); (ii) cardiovascular disease (CVD) clinically evident in the last 6 months, manifested by a myocardial infarction or a stroke; (iii) a neurological, muscular, or rheumatologic disease that difficulte the position of the individual on the vibrating platform; (iv) disabling clinical disease according to the evaluation; (v) speech therapy or respiratory physiotherapy in the last 3 months; (vi) body mass index (BMI) > 40 kg/m$^2$; (vii) orthodontic therapy with intraoral device; (viii) individuals who refuse to sign the consent form.

The participants declared that they were sedentary. They were instructed to continue their daily activities, dietary habits, and medications during the period of the investigation. The drugs used by the participants of both groups were diuretics, beta blockers, calcium channel blockers, angiotensin converting enzyme inhibitors, and angiotensin receptor antagonists.

*2.2. Interventions*

### 2.2.1. Fixed Frequency Group (Control Group)

After the randomization, nine individuals (seven women and two men) allocated in the FFG performed a protocol twice a week, for 6 weeks. In this protocol, the individuals were positioned in a squat position, barefoot and with 130° knee flexion (Figure 1A). The biomechanical parameters were 2.5, 5 and 7.5 mm (Figure 1B–D) of PPD and 5 Hz for 10 s of vibration and 110 s of non-vibration in each bout. From 1 to 4 weeks, were performed 3 bouts in each session, totaling 18 min of total time. From 5 to 8 weeks, were performed 4 bouts in each session, totaling 24 min of total time. From 9 to 12 weeks, were performed 5 bouts in each session, totaling 30 min of total time. The participants performed dynamic and static squats in the sessions.

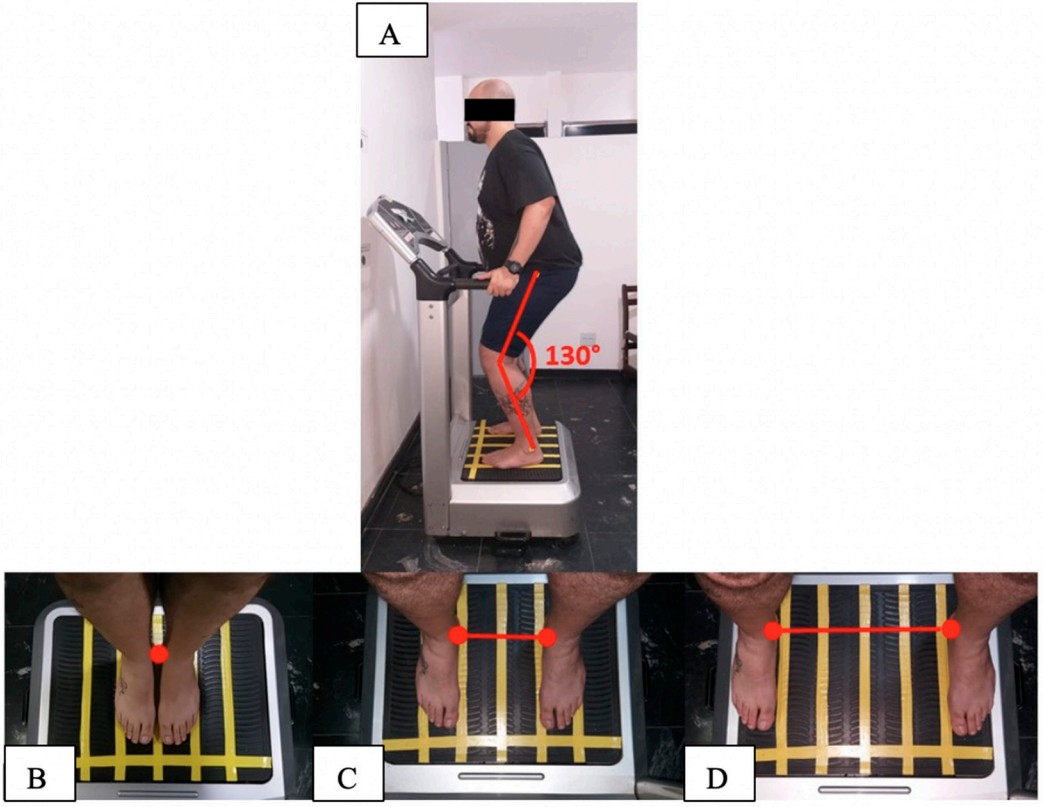

**Figure 1.** Biomechanical parameters during the intervention (fixed frequency (FF) and variable frequency (VF) groups) using the side alternating vibratory platform; (**A**) individual in a stand position, barefoot and with 130° knee flexion; (**B**) individual in 2.5 mm of peak-to-peak displacement (PPD) on the base of the vibrating platform with medial malleoli together; (**C**) individual in 5 mm of PPD on the base of the vibrating platform 21 cm apart between the medial malleoli; (**D**) individual in 7.5 mm of PPD on the base of the vibrating platform 38 cm apart between the medial malleoli.

### 2.2.2. Variable Frequency Group (WBVE Group)

After the randomization, ten individuals (eight women and two men) allocated in the VFG performed the protocol twice a week, for 6 weeks. In this protocol, the individuals were positioned in

a squat position, barefoot and with 130° knee flexion (Figure 1) The biomechanical parameters of the MV were 2.5, 5 and 7.5 mm (Figure 1B–D) of the PPD; the frequency was 5 Hz in the first session and it increased by one Hz in each session, and it was 16 Hz in the last session. During the intervention, were performed 60 s of MV and 60 s of non-vibration in each bout. From 1 to 4 weeks were performed 3 bouts in each session, for a total time of 18 min. From 5 to 8 weeks, were performed 4 bouts in each session, for a total time of 24 min. From 9 to 12 weeks were performed 5 bouts in each session, for a total time of 30 min. Dynamic and static squats were performed in interspersed sessions.

### 2.3. Anthropometric Measurements and Physiological Parameters

All of the measurements were performed in the same location, at room temperature. Height and body mass were measured on a digital balance (MIC 200 PPA, Micheletti, São Paulo, Brazil). The BMI was calculated by dividing body mass (in kg) by square of height (in m) [34]. WC was obtained with a flexible measuring tape connecting the midpoints between the last costal arch and iliac crest, at the end of the soft exhalation and in the orthostatic position [34]. NC was obtained using a flexible metric tape just below the prominence of the larynx [35].

An automated device (OMRON, model HEM-7113, Dalian, China) was utilized to record the SBP and DBP (mm Hg), and the HR (beats/min) was measured on the left arm of seated patient after a 10-min rest. Three measurements were performed with 1 min of rest after each measurement. The mean of these 3 records of SBP and DBP and HR was used in the analyses of the WBV exercise group and the control group [36]. These evaluations were performed before and after the 6-week intervention in both groups.

The data of these parameters were expressed in mean and standard derivation and the Wilcoxon rank test was performed.

### 2.4. Questionnaires

The PSQI [15] was used to assess sleep quality and disturbances over a 1-month period. This questionnaire is composed of 19 self-rated questions and 5 questions that should be answered by bedmates or roommates. The 19 questions are separated into 7 categories and there is a score from 0 to 3. The components of this questionnaire are subjective sleep quality, sleep latency, sleep duration, habitual sleep efficiency, sleep disturbances, use of medication, and daytime dysfunction. The maximum PSQI score is 21 points. Higher scores indicate poor quality of sleep [37].

The ESS [16] is composed of 8 self-reported questions and it is used to assess the self-reported level of daytime sleepiness [38]. These items have a four-point scale (0 indicating "would never nod off" and 3 indicating a "strong chance of nodding off"). The questionnaire questions are related to daily activities. The questionnaire score ranges between 0 and 24. Higher total scores are related to more sleepiness [39].

The Berlin Questionnaire [17] has been proposed as a tool in screening OSA and validated in primary care. The subjects were classified into high risk (if there were 2 or more categories where the score was positive), and low risk (if there was only 1 or no categories where the score was positive). The percentage os individuals with high risk to OSA was calculated in both groups [40,41].

The PSQI, ESS, and Berlin Questionnaire were only measured before and after the 6-week intervention in both groups.

### 2.5. Statistical Analyses

For the sample size calculation, we used a standard deviation of 3.1, a margin of error of 2, the significance level of 5%, and a sample size of 9 individuals in each group, considering the PSQI [42].

The Wilcoxon signed-rank test with continuity correction was used in the analyses of the DBP, SBP, HR, WC, NC, the PSQI, and ESS. Descriptive statistics, median, and interquartile range (IQR) are also reported. Wilcoxon signed-rank test is a nonparametric test that was used to determine if paired samples, before and after intervention, have the same distribution. Fisher's exact test was

applied to the qualitative variable, Berlin, to asses if the proportion of people in each class (high risk and low risk) have been affected by the experiment. In the analyses of the PSQI and ESS, the values are expressed in scores. The statistical analyses of the Berlin Questionnaire considered only the percentage of individuals with a high risk to OSA.

All statistical analysis was performed using R software, version 3.5.0 [43] and the R Librarie Table 1 [44]. Results are considered statistically significant if the *p*-value is under 0.05 ($p \leq 0.05$).

## 3. Results

The flow diagram with the enrolment of the study is shown in Figure 2. Thirty-one individuals were recruited, two individuals were excluded because they did not have risk of sleep apnea, three declined to participate, and seven individuals declined due to other reasons (economical problem related to the transportation).

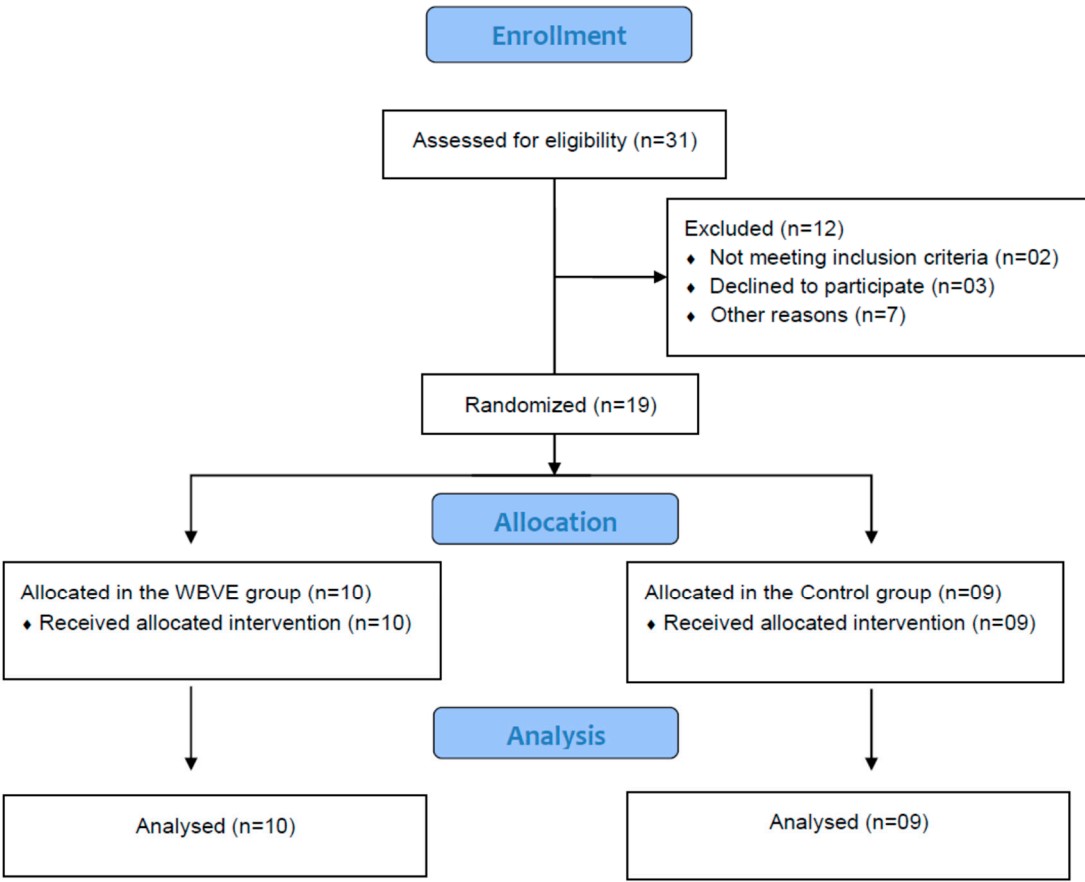

**Figure 2.** Flow diagram of the intervention.

Table 1 shows some anthropometric parameters of the individuals that were exposed to WBV exercise (FV group) and to only 5 Hz (FF group). An analysis is shown of both groups before and just after the treatments. It was verified that the BMI, NC, SBP, and DBP ($p > 0.05$) were not significantly altered in both groups. However, a significant increase in the HR ($p < 0.05$) was found in both groups exposed to mechanical vibration, only 5 Hz (FFG) and variable frequency 5–16Hz (VFG). Furthermore, a reduction was found in the group that was submitted to the variable frequency 5–16 Hz (VFG) in the WC.

**Table 1.** Anthropometrical parameters of the individuals exposed to variable frequency group (intervention 5–16 Hz) and to only 5 Hz (fixed frequency group).

| | FFG before (Median [IQR]) | FFG after (Median [IQR]) | *p* | VFG before (Median [IQR]) | VFG after (Median [IQR]) | *p* |
|---|---|---|---|---|---|---|
| BMI (kg/m$^2$) | 29.50 [29.0, 34.20] | 29.60 [28.90, 34.10] | 0.54 | 35.40 [32.40, 36.60] | 34.70 [32.88, 36.08] | 0.23 |
| WC (cm) | 105.00 [101.20, 106.30] | 101.00 [100.00, 105.00] | 0.23 | 108.90 [103.75, 116.10] | 107.00 [102.17, 112.85] | 0.02 |
| NC (cm) | 38.00 [36.40, 38.50] | 36.60 [35.60, 39.50] | 0.29 | 38.00 [37.58, 38.88] | 38.30 [37.17, 39.00] | 0.25 |
| SBP (mm Hg) | 129.00 [128.00, 130.00] | 127.00 [121.00, 128.00] | 0.12 | 130.00 [122.50, 132.50] | 130.00 [120.00, 132.00] | 0.76 |
| DBP (mm Hg) | 78.00 [75.00, 83.00] | 75.00 [68.00, 87.00] | 1.00 | 74.00 [72.25, 75.75] | 70.00 [60.75, 74.75] | 0.18 |
| HR (bpm) | 57.00 [56.00, 62.00] | 62.00 [60.00, 66.00] | 0.05 | 63.50 [59.75, 66.00] | 72.00 [61.25, 83.00] | 0.05 |

BMI—body mass index, WC—waist circumference, NC—neck circumference, SBP—systolic blood pressure, DBP—diastolic blood pressure, HR—heart rate, FF—fixed frequency group, VF—variable frequency group.

Figure 3 shows the values of the scores of the PSQI of the individuals of the group exposed only to mechanical vibration of 5 Hz (FFG-control). A significant decrease was found (p = 0.01) of the score (after and before) of the individuals of the FF group (exposure to 5 Hz).

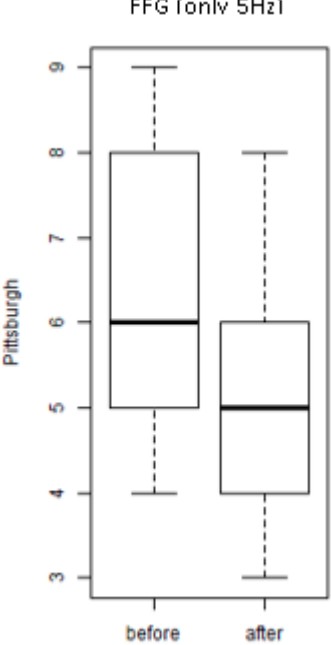

**Figure 3.** Scores of the Pittsburgh Sleep Quality Index (PSQI) of the individuals of the group exposed only to a mechanical vibration of 5 Hz (FFG) (before and after).

Figure 4 shows the values of the scores of the PSQI of the individuals of the groups exposed to the treatment with WBV exercises (VFG). It is observed that an important and significant (*p* = 0.008) decrease of score in the group after the exposure to various frequencies (intervention) was seen. This finding indicates an improvement of quality of the sleep.

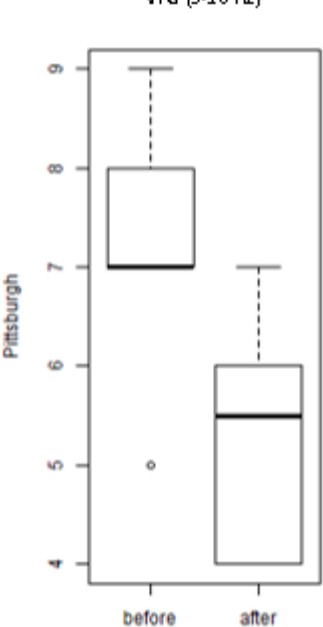

**Figure 4.** Scores of the PSQI of the individuals exposed to the whole-body vibration (WBV) intervention with variable frequency (VFG).

Figure 5 shows the values of the scores of the ESS of the individuals of the group exposed only to mechanical vibration of 5 Hz (FFG). It is observed that there was no significant decrease ($p = 1.0$) of the score (after and before) of the individuals. This finding indicates no reduction of daytime sleepiness.

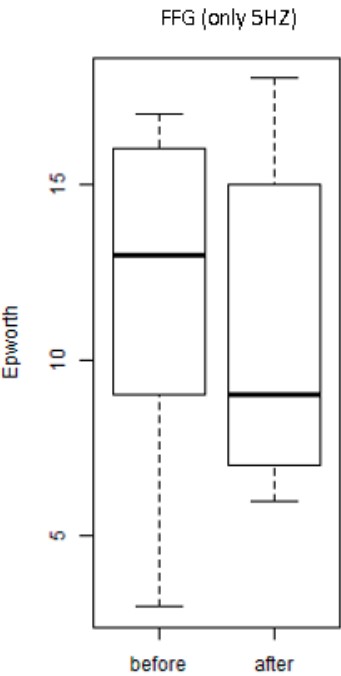

**Figure 5.** Scores of the Epworth Sleepiness Scale (ESS) of the individuals of the SG individuals of the FFG (before and after).

Figure 6 shows the values (before and after the interventions) of the scores of the ESS scale of the individuals of the groups exposed to the treatment with WBV exercises (VFG). It is observed that there

was a significant ($p = 0.04$) decrease of the score in the group after the exposure to various frequencies (intervention). This finding indicates a reduction of daytime sleepiness.

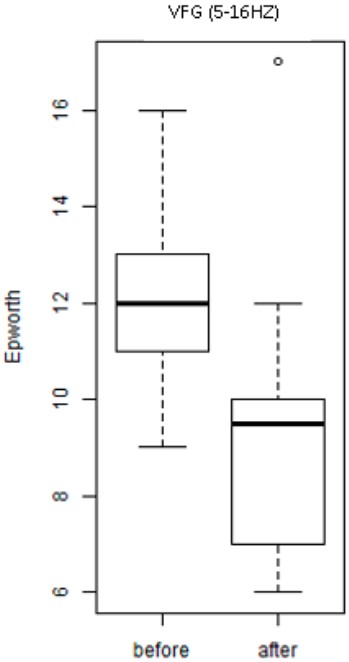

**Figure 6.** Scores of the ESS of the individuals of the group exposed to the intervention with WBV exercises (VFG).

Considering the Berlin Questionnaire, no alteration in ter percentage of individuals with risk to OSA were significantly found in both groups, the FF $p = 0.29$ and FVG $p = 1.00$.

## 4. Discussion

The underlying etiology of MetS is multifactorial, however, they are not well known, but sedentary lifestyles and unhealthy dietary habits favor the appearance of MetS [45]. Considering the high prevalence and strong and undesirable complications, early identifying and controlling the modified risk factors are key prevention methods to counter the development of MetS and its progression to CVD.

It is relevant to point out that, to our knowledge, this is the first work that was performed to evaluate the effect of WBV exercise in sleep quality of MetS individuals. As it was hypothesized that the WBV exercise would be able to improve the sleep quality of MetS individuals, the quality of sleep was improved (PSQI) and a reduction of daytime sleepiness (ESS) in the MetS was found due to a protocol with variable frequencies (5–16 Hz) of WBV. A protocol with fixed frequency (5 Hz) also improved the quality of sleep (PSQI).

It was shown (Table 1) that the 6-week intervention with WBV exercise with variable frequencies (from 5 up to 16 Hz) (VF group) did not alter significantly the BMI, NC, SBP, and DBP. The same finding was verified in the individuals with fixed frequency (5 Hz). These findings agree with other studies. Feairheller et al. [46], in a study of six months, have shown that aerobic exercise training is enough to elicit improvements in vascular structure and function in African Americans, even without alterations on BP measurements (SBP and DBP) in African Americans. Moreover, Sá-Caputo et al. [31] have reported that, in individuals with MetS exposed to acute WBV exercise, the SBP and DBP were not altered. Figueroa et al. [21] have reported that the BMI of overweight/obese women was not modified after 6 weeks of WBV exercises.

Considering the HR (Table 1), an increase was verified in FFG (only 5 Hz) and VFG (variable 5–16 Hz), and this is in agreement with Sunita et al. [47] that have also observed an increase of the HR

in response to exercise. Moreover, Kang et al. [48] have described an association between resting HR and cardiorespiratory fitness with an elevated resting HR.

The findings presented in Table 1 indicated no alterations on SBP and DBP due to WBVE (a possible stressor), however, Massar et al. [6] reported that individuals with poor habitual sleep efficiency during the week before stress induction (Trier Social Stress Test) responded with higher stress-related elevations of blood pressure and cortisol levels as compared to subjects with high sleep efficiency.

In the current study, no alterations on the NC were found. However, a decrease on the WC was found (Table 1). Sîrbu et al. [49] have compared some anthropometric parameters in trained and sedentary healthy young students. It was found that in moderately aerobic trained students, the BMI and BP were not significantly different between the two groups. However, a significantly lower NC was found in the trained students. Considering the WC, in agreement with our results, King et al. [50] have verified that a 12-week supervised aerobic exercise intervention has reduced significantly the WC of overweight/obese men and women. But, Sîrbu et al. [49], have not found alteration in the WC due to exercise.

WBV exercise is an intervention with several improvements in populations with various diseases [23–25,31]. In this current study, it was verified that WBV exercise has improved the quality of the sleep in MetS when we used the PSQI (Figures 3 and 4) and ESS (Figure 6). This finding is partially in agreement with Souza et al. [51] who have evaluated, in a population of 16 individuals with OSA, the effectiveness of inspiratory muscle training (IMT) on sleep and functional capacity to exercise. It was a significant improvement in sleep quality was seen with PSQI values, but no significant changes were seen in daytime sleepiness (ESS) after the intervention. Furthermore, Reid et al. [52] have verified that moderate aerobic PE improved the sleep quality of sedentary adults considering the PSQI and the ESS. Brandão et al. [53] have reported that home exercise improves the quality of sleep and daytime sleepiness of elderlies considering, respectively, the PSQI and the ESS.

The findings of the current work have verified that the sleep quality was not altered when the Berlin Questionnaire was used. This result agrees with Brandão et al. [53] who have reported that home exercise does not alter the risk to OSA when the same questionnaire was used.

Kline et al. [54], evaluated the utility of PE for improving daytime functioning in adults with OSA. Sleepiness and functional impairment due to sleepiness also were improved following exercise versus control, though these changes were not statistically significant. It is suggested that PE may be helpful for improving aspects of daytime functioning of adults with OSA. Moreover, Yilmaz Gokmen et al. [55] have investigated the effects of Tai Chi and Qigong training on severity of OSA for 12-weeks. In the intervention group, there was a statistically significant decrease in the ESS. However, Itoh et al. [56], in a cross-sectional study among non-obese male workers in Japan, found no significant association between physical activity and the risk of sleep-disordered breathing.

Considering the MetS, not all forms of exercise are equally effective and safe; although aerobic exercise [57] or resistance training have been associated with decreased cardiovascular disease risk factors, obesity, or MetS severity [58,59]. Due to the pain or even the low physical fitness, most individuals are unable or unwilling to perform these exercises [60]. Given the limitations for some types of exercises that many individuals report, different forms of exercise are being suggested for MetS individuals [61]. WBV exercise has improved several parameters in MetS individuals [26,31,32] and in the current study a decrease of the WC was found (Table 1).

This study has some limitations, no control of daily activity, daily working, lifestyle habits, smoking, PE, and daily energy intake. The follow-up data were not available after the intervention. The external validity of this intervention considering its generalizability to other settings (like the everyday-living condition) was not explored and only a small sample was used. The big ratio of female to male would be also considered as a limitation of the study. Moreover, a sham group and a group of individuals without MetS were not used. Furthermore, the analysis of the questionnaires was not done in the weeks in which there were changes of the frequency and duration of the intervention in the VFG. Some points of the questionnaires are influenced by factors that are not taken into account in the current

study, as categories of individuals (good-sleepers and bad-sleepers), sleep duration, sleep inertia, and sleep latency. In consequence, further investigations are required. However, putting together all the considerations, the strength was to verify that a WBV exercise might be suggested as an option to the management of MetS individuals with some important outcomes, and it would be expected that a WBV exercise might lead to responses that can reflect in improvements of the sleep quality.

## 5. Conclusions

In conclusion, WBV intervention was capable of interfering with physiological mechanisms with effects on the WC and HR, leading to the improvement of the quality of sleep in MetS individuals. Although the calculated sample size determined a study with a small number of individuals, WBV exercise might be an important clinical intervention to the management of some factors associated with poor quality of sleep (FFG and VFG) and in the daytime sleepiness in MetS individuals with variable frequencies (5–16Hz) (VFG). However, further investigations are necessary to try to understand the mechanisms that underlines this positive effect of the WBV exercise in the studied population, as well as with individuals with sleep disturbances without MetS.

**Author Contributions:** Conceptualization, R.T., A.S. (Alessandro Sartorio), M.B.-F., and D.d.C.d.C.D.S.-C.; data curation, C.F.A., J.A.B., V.L.X., and D.L.B.; formal analysis, J.A.B., V.L.X., D.L.B., A.C.L., V.A.M., A.S. (Anelise Sonza), and R.T.; funding acquisition, M.B.-F.; Investigation, C.F.A., P.d.C.d.P., L.A., A.R.d.S., A.F.-S., L.L.P.-D., A.L.P.d.S., C.L.B.-O., J.P.-F., M.C.M.-F., R.G.M., Y.T.-S., E.M.-M., E.d.O.G.-A., B.B.M.d.O., M.F.N., and L.F.F.-S.; methodology, C.F.A., P.d.C.d.P., L.A., A.R.D.S., A.F.-S., L.L.P.-D., A.L.P.d.S., C.L.B.-O., J.P.-F., M.C.M.-F., R.G.M., J.A.B., Y.T.-S., E.M.-M., E.d.O.G.-A., B.B.M.d.O., M.F.N., L.F.F.-S., and A.S.(Alessandro Sartorio); project administration, M.B.-F. and D.d.C.d.C.D.S.-C.; resources, C.F.A., P.d.C.d.P., L.A., A.R.d.S., R.G.M., and B.B.M.d.O.; software, V.L.X.; supervision, M.B.-F. and D.d.C.d.C.D.S.-C.; validation, V.L.X., M.B.-F., and D.d.C.d.C.D.S.-C.; visualization, A.S.(Anelise Sonza) and R.T.; writing—original draft, C.F.A.; Writing—review and editing, A.C.L., V.A.M., A.S. (Anelise Sonza), A.S. (Alessandro Sartorio), M.B.-F., and D.d.C.d.C.D.S.-C.

**Funding:** This study was supported in part by the *Coordenação de Aperfeiçoamento de Pessoal de Nível Superior - Brazil (CAPES) - Finance Code 001, the *Conselho Nacional de Desenvolvimento Científico e Tecnológico (CNPq) and the Fundação de Amparo à Pesquisa do Estado do Rio de Janeiro* (FAPERJ).

**Conflicts of Interest:** There is no conflict of interest.

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
