# Peer review of "Effects of Whole-Body Vibration Exercises on Parameters Related to the Sleep Quality in Metabolic Syndrome Individuals: A Clinical Trial Study"

_applsci, doi:10.3390/app9235183_

Round 1
Reviewer 1 Report
Manuscript ID: applsci-636826
Title: Effects of whole-body vibration exercises on parameters related to the
sleep quality in metabolic syndrome individuals: a clinical trial study
General comment
A serious drawback of the study is the lack of a control group (persons who do not have MetS), so the authors ' conclusions regarding the effectiveness of the procedures used in relation to persons with MetS have no evidence. It is possible that the procedures have an effect on individuals with poor night sleep quality and daytime sleepiness, regardless of whether or not they have signs of MetS. The sample size is too small, so conclusions should be formulated very carefully. Co-authors of the study are 25 people, this is more than the number of participants in the study. The question arises about the contribution of each of the co-authors to the study. Statistical analysis of the study results is not reliable enough. The indicators under consideration are influenced by a very large number of related factors that are not taken into account. PSQI allows you to distinguish the categories of persons: good-sleepers and bad-sleepers. The authors need to analyze the impact of the procedures used on the frequency of detection of these groups. The authors need to analyze in which PSQI sub-scales the greatest changes occurred as a result of the procedures (sleep duration, sleep inertia, sleep latency, etc.).Summary
The number and sex composition of the study participants are not indicated.Introduction
Line 76-77. The phrase sounds ambiguous: "high quality sleep in people with MetS"? Line 78. ... .. [5,6]. Massar et al. [7] ... Line 83. This is a repeat, you must combine with the phrase Line 76-77. Line 90. This is a repeat, you must combine with the phrase Line 76-77. Line 96. Refine the phrase, what are these indicators used for? At the end of the introduction, it is necessary to set the goals and objectives of the study, to formulate hypothesis.Methods
Line 115. The number and sex composition of the study participants are not indicated. Line 117-119. It is necessary to indicate the criteria by which MetS was diagnosed (criteria for the inclusion of patients in the study). Line 194. More daytime sleepiness ...Results
Table 1. The difference in WC 1.9 cm is too small; this result does not make sense in the conclusions and in the summary. Table 1. A significant increase in heart rate was noted, but this is not mentioned in the Results section.Discussion
Line 279-282. It is necessary to discuss whether this is the positive or negative effect of the procedures on the health of individuals with MetS. Line 330-332. Since the study did not involve individuals without MetS, this claim has not been proven. The procedure can apply to people with poor quality of sleep, regardless of the presence or absence of MetS. See comment 5.Conclusion
In the conclusion section it is inadmissible to use the word “possible”. Only the main results should be indicated.References
In many cases, the volume and page are not indicated.Author Response
Reviewer 1 –
Yes |
Can be improved |
Must be improved |
Not applicable |
|
Does the introduction provide sufficient background and include all relevant references? |
( ) |
( ) |
(x) |
( ) |
Is the research design appropriate? |
( ) |
( ) |
(x) |
( ) |
Are the methods adequately described? |
( ) |
( ) |
(x) |
( ) |
Are the results clearly presented? |
( ) |
( ) |
(x) |
( ) |
Are the conclusions supported by the results? |
( ) |
( ) |
(x) |
( ) |
We improved the introduction, the research design, the methods, the results and the conclusion.
The number and sex composition of the study participants are not indicated.
It was rewrote and we added the solicited information (line 51 and 52).
Introduction
Line 76-77. The phrase sounds ambiguous: "high quality sleep in people with MetS"?
It was rewrote.
Line 78. ... .. [5,6]. Massar et al. [7] ...
It was rewrote.
Line 83. This is a repeat, you must combine with the phrase.
It was rewrote.
Line 76-77. Line 90. This is a repeat, you must combine with the phrase
It was rewrote.
Line 76-77. Line 96. Refine the phrase, what are these indicators used for? At the end of the introduction, it is necessary to set the goals and objectives of the study, to formulate hypothesis.
It was rewrote and we added the solicited information (line 124 – 128).
Methods
Line 115. The number and sex composition of the study participants are not indicated.
It was rewrote and we added the solicited information (line 159 and 169).
Line 117-119. It is necessary to indicate the criteria by which MetS was diagnosed (criteria for the inclusion of patients in the study).
It was rewrote and we added the solicited information (line 145).
Line 194. More daytime sleepiness ...
It was rewrote.
Results
Table 1. The difference in WC 1.9 cm is too small; this result does not make sense in the conclusions and in the summary. Table 1. A significant increase in heart rate was noted, but this is not mentioned in the Results section.
It was rewrote and we added the solicited information.
Discussion
Line 279-282. It is necessary to discuss whether this is the positive or negative effect of the procedures on the health of individuals with MetS. Line 330-332. Since the study did not involve individuals without MetS, this claim has not been proven. The procedure can apply to people with poor quality of sleep, regardless of the presence or absence of MetS. See comment 5.
It was rewrote and we added the solicited information.
Conclusion
In the conclusion section it is inadmissible to use the word “possible”. Only the main results should be indicated.
It was rewrote and we added the solicited information.
References
In many cases, the volume and page are not indicated.
We added the solicited information.

Reviewer 2 Report
This clinical trial investigated the effect of Whole-body vibration (WBV) exercise on parameters related to the sleep quality in Metabolic syndrome (MetS) individuals. The authors concluded that WBV exercise might an important clinical intervention to the management of some factors associated with poor quality of sleep and in the daytime sleepiness in MetS individuals with VF (5-14Hz). Probably, WBV exercise was capable in interfering with physiological mechanisms with effects on the waist circumference, leading to the improvement of the quality of sleep in MetS individuals.
It is an interesting work, but I would like to make some comments:
My main concern is the small number of patients involved in the study (n=10 and n=9 ). The patients of the VFG group had higher BMI. This, with the small number of patients may have implicated the results Where there any differences between genders as the waist to hip ratio may be different between genders. Where there differences between medications used in the 2 groups? The sleep of the patients was not assessed by a sleep study, but just with questionnaires. If the analysis was done by the weeks of changing the duration of intervention were there any differences?Author Response
Reviewer 2-
Yes |
Can be improved |
Must be improved |
Not applicable |
|
Does the introduction provide sufficient background and include all relevant references? |
(x) |
( ) |
( ) |
( ) |
Is the research design appropriate? |
( ) |
(x) |
( ) |
( ) |
Are the methods adequately described? |
(x) |
( ) |
( ) |
( ) |
Are the results clearly presented? |
(x) |
( ) |
( ) |
( ) |
Are the conclusions supported by the results? |
( ) |
(x) |
( ) |
( ) |
We improved all the sessions solicited.
Comments and Suggestions for Authors
My main concern is the small number of patients involved in the study (n=10 and n=9). The patients of the VFG group had higher BMI. This, with the small number of patients may have implicated the results Where there any differences between genders as the waist to hip ratio may be different between genders.
It was rewrote and we added the solicited information.
Where there differences between medications used in the 2 groups?
It was rewrote and we added the solicited information.
The sleep of the patients was not assessed by a sleep study, but just with questionnaires. If the analysis was done by the weeks of changing the duration of intervention were there any differences?
It was rewrote and we added the solicited information (limitations session). (line 153 – 156 and 356 – 369).

Round 2
Reviewer 1 Report
No comments.